# How Cognition Influences Chinese Residents’ Continuous Purchasing Intention of Prepared Dishes under the Distributed Cognitive Perspective

**DOI:** 10.3390/foods13162598

**Published:** 2024-08-20

**Authors:** Yuelin Fu, Weihua Zhang, Ranran Wang, Jiaqiang Zheng

**Affiliations:** College of Economics and Management, Shanghai Ocean University, Shanghai 201306, China; 18715850585@163.com (Y.F.); rrw1512@163.com (R.W.); zhengjq365@outlook.com (J.Z.)

**Keywords:** prepared dishes, risk perception, distributed cognition theory, continuous purchasing intention

## Abstract

Enhancing residents’ purchasing intention of prepared dishes is crucial for the sustainable development of the prepared dishes industry. Understanding how residents’ cognition influences their continuous purchasing intention can provide valuable insight for developing and refining company strategies, thereby reducing industry development obstacles. Based on the theory of distributed cognition, this study utilizes questionnaire data from urban residents in Beijing and Shanghai, and employs Structural Equation Modeling to explore the influence of cognition on the continuous purchasing intention of Chinese urban residents towards prepared dishes. The study results reveal that: (1) Individual power and geographical power have a significant positive effect on residents’ continuous purchasing intention for prepared dishes, while cultural power does not have a significant effect. (2) Risk perception partially mediates the effect of individual power and geographical power on continuous purchasing intention and fully mediates the effect of cultural power on continuous purchasing intention. Recommendations include: (1) The government should enhance standardization and supervision to create a favorable consumption environment; (2) Enterprises should provide more objective and transparent information to improve residents’ knowledge of prepared dishes and establish a good reputation.

## 1. Introduction

With the continuous development of the economy and society and the accelerated pace of life, residents’ eating habits and dietary structures are constantly evolving. Prepared dishes, as a kind of convenient and fast food, are increasingly gaining people’s attention. In 2023, the No. 1 Central Document of China explicitly proposed the need to “cultivate and develop the prepared dishes industry” for the first time, providing a strong impetus for the industry’s growth. In March 2024, China’s General Administration of Market Supervision, along with six other ministries and commissions, issued the “Notice on Strengthening Food Safety Supervision of Prepared Dishes and Promoting High-quality Development of the Industry” [1]. This notice clearly defines the scope and definition of prepared dishes, mandates the advancement of regulatory and standard systems for prepared dishes, and provides explicit guidelines for the high-quality development of the industry. Driven by both consumer demand and strong national policy support, the prepared dishes industry has entered a rapid growth phase. According to enterprises search data [2], by the end of 2023, China had approximately 61,900 enterprises related to prepared dishes, with an output value exceeding 500 billion yuan, and it is expected to exceed one trillion yuan in 2026. While the development of China’s prepared dishes market has shown positive trends, the industry faces significant challenges. For instance, the prepared dishes industry has been exposed as “one of China’s top ten consumption chaos in 2022”, and food safety and other issues have become an urgent need to be solved in the prepared dishes industry. Chinese urban residents currently have concerns about the safety and health of prepared dishes [3]. Therefore, their continuous purchasing intention of prepared dishes is relatively low. This lack of consumer recognition and acceptance of prepared dishes hinders the expansion and development of the prepared dishes market and poses a threat to the sustainable development of the entire industry.

Cognition is how well residents know about a product and can directly or indirectly influence their continuous purchasing intention. The more information and knowledge consumers have about a product, the stronger their continuous purchasing intention [4]. For genetically modified foods [5,6], farmers’ demand for pollution-free pesticides [4], and commercial insurance [7], similar conclusions have been reached. Additionally, cognition can indirectly influence consumer purchasing intention through mediating variables. Empirical studies on the purchasing intention of genetically modified foods [8] and fresh agricultural products [9] have found that the more individuals understand the product, the lower their perceived risk and the stronger their purchasing intention.

Currently, there is extensive international research on prepared foods, particularly in high-income countries. Internationally, prepared dishes are classified under the fourth category of the NOVA classification of ultra-processed foods (UPFs) [10]. International studies on ultra-processed foods, encompassing countries like the United States, Brazil, Mexico, Europe countries, India, South Korea, Singapore, and Fiji, have focused on the association between the consumption of UPF and individual health issues like obesity [11] and hypertension [12], as well as the assessment of the consumption of UPF in relation to the individual nutritional [13] and dietary quality [14,15]. Additionally, there are studies on purchasing differences of UPFs among consumers with varying socio-economic backgrounds based on cross-sectional observational studies [16,17], the relationship between psychological stress and UPF consumption in adults [18], and factors influencing UPF consumption among young people [19]. Other areas of research include consumption trends [20], consumption patterns [21], and food labeling information and purchasing intention [22]. While UPFs consumption is widespread globally, there is a paucity of research on purchasing intention related to UPFs in various countries.

Current research in China on prepared dishes primarily focuses on the levels of industry development, technology research, industry standardization, etc. The research at the level of industry development focuses on the current status and trends of the entire prepared dishes industry [23,24], as well as regional or special cuisines [25,26]. The technical aspects of the research mainly focus on the processing of prepared dishes [27], packaging [28], storage and transportation [29], safety control [30], and other technologies exploration [31,32,33]. Research at the level of industry regulation mainly focuses on the traceability system for prepared dishes [34], food safety and legal regulation aspects [35,36], regulatory proposals for food additives [37], and other studies. In the prepared dishes consumption studies, some scholars have studied the factors influencing consumers’ purchasing intention of prepared dishes [38], the roles of perceived risk and trust on consumers’ continuous purchasing intention for prepared dishes [39] and ultra-processed food consumption and risk of obesity in Chinese adults [40]. Existing research on prepared dishes has rarely explored the relationship between residents’ cognition and their continuous purchasing intention of prepared dishes. From a consumer research perspective, empirical research on risk perception and continuous purchasing intention has already been applied in the area of organic food [41], genetically modified (GM) food [42], and green food [43]. Therefore, it indicates a gap in empirical research on this topic.

Therefore, based on the distributed cognition theory, this paper introduces the concept of risk perception and employs Structural Equation Modeling to deeply analyze urban residents’ cognition and the impact of cognition on continuous purchasing intention of prepared dishes, using data coming from urban residents in Beijing and Shanghai. Specifically, the study addresses the following questions: 

(1) Does the cognition of prepared dishes among Chinese urban residents influence their continuous purchasing intention? How do different dimensions of cognition influence their continuous purchasing intention?

(2) Does risk perception play a mediating role between dimensions of cognition and continuous purchasing intention? Additionally, how does it play a mediating role?

Based on distributed cognition theory and questionnaire data from residents of Beijing and Shanghai, China, this paper utilizes the structural equation model and mediating effects methods to comprehensively explore the effects of cognition and its subdimensions on residents’ continuous purchasing intention and to examine the mediating effects of risk perception on the relationship between cognition and residents’ purchasing intention for prepared dishes. This study extends the scope of the application of distributed cognition theory and analyzes the cognition of Chinese urban residents towards prepared dishes from the dimensions of individual power, geographical power and cultural power. It also examines the impact of each dimension of cognition on residents’ continuous purchasing intention of prepared dishes, enriches the research on the consumption behavior related to prepared dishes and provides a reference for applying distributed cognition theory to study the purchasing intention and consumption behavior.

The rest of the paper is organized as follows: Section 2 presents the research hypotheses, Section 3 presents the methods and data, Section 4 presents the research results and analysis, Section 5 discusses the findings and contributions, Section 6 presents the conclusions and managerial implications, and Section 7 outlines the limitations and future research.

## 2. Research Hypotheses

Distributed Cognition Theory is derived from psychological research about individual cognition. It suggests that cognition is formed by the interaction of the individual with the external environment, emphasizing that the individual maintains their independence while interacting with the environment. The concentric circles model proposed by Hatch and Gardner states that the model includes individual power, geographical power, and cultural power from the center outward [44]. Individual power mainly emphasizes the influence of personal subjective initiative and preference on cognition, while geographical power and cultural power focus on external environmental factors affecting cognition. Based on this theory, this paper divides cognition into three dimensions, including individual power, geographical power, and cultural power, and then explores their different effects on urban residents’ continuous purchasing intention of prepared dishes, respectively. Figure 1 presents the structural equation model, illustrating the relationship among variables: individual power, geographical power, cultural power, risk perception, and continuous purchasing intention.

Individual power cognition refers to the knowledge that is acquired from personal experiences and eating preferences [35]. For consumers, relevant food purchasing experiences have a significant impact on individual cognition [45]. When residents seek product information, they utilize search engines, shopping websites, and personal experiences. These methods effectively generate individual cognition and provide comprehensive product knowledge [46]. Moreover, the richer the personal purchasing experience, the more knowledge about the product is accumulated, the greater the role played by trust and knowledge, and the greater the impact on the future purchasing intention and behavior of green food [43]. By increasing the purchasing experience of prepared dishes, residents have a better understanding of the convenience, safety, and nutritional value of prepared dishes, which enhances their perceptions of improved quality of life and increases product utility, which in turn can increase consumers’ continuous purchasing intention. Based on this, the following hypothesis is proposed:

**H1.** *Individual power has a significant positive effect on urban residents’ continuous purchasing intention of prepared dishes*.

Geographical power cognition refers to the resources an individual possesses and the perceptions formed under the influence of people who have an impact on their behavior [47]. Geographical power in the field of food consumption includes personal capabilities (including time, money, experience, and other factors [48]), skills, accessible information, channel accessibility, other people’s evaluation and other factors [49,50]. Peer effect points out that an individual’s behavior will be influenced by the behavior of other people around them with the same status [51]. Residents also can be influenced by changes in the consumption trends of people around them. When they pay more attention to the quality and safety of food, they are more inclined to choose green and healthy food [52,53]. Witek and Kuźniar [43] have found that the opinions of people around them, such as family members, partners, friends, or peers, have a very important influence on an individual’s own purchasing behavior. Based on this, the following hypothesis is proposed:

**H2.** *Geographical power has a significant positive effect on urban residents’ continuous purchasing intention of prepared dishes*.

Cultural power cognition refers to the understanding formed by individuals under the influence of intangible aspects of their environment, such as culture or institutions [54]. In China, the majority of prepared dish producers are small and medium-sized enterprises or individual businesses, which often have lower levels of product standardization, leading to potential food safety issues and product quality risks. To strengthen food safety supervision and promote the healthy development of the prepared dishes industry, Chinese ministries, and local governments have introduced multiple policies. These policies support the industry’s healthy growth, increase consumers’ cognition of government regulatory systems and processes, and enhance consumers’ confidence in prepared dishes, thereby boosting their continuous purchasing intention. Han and Zhang [55] found that the government regulation status and the level of information disclosure of traceable foods significantly affect consumer purchasing intention. Zhuang and Yu [56] found that ethically induced negative product performance-based exposure events have a significant negative effect on consumers’ purchasing intention for those with low cognitive demand. Based on these findings, the following hypothesis is proposed:

**H3.** *Cultural power has a significant positive effect on urban residents’ continuous purchasing intention of prepared dishes*.

Risk perception is the uncertainty and unfavorable consequences that consumers feel when purchasing a product or service [57]. Consumers’ risk perception of prepared dishes refers to their judgment of the risk level of quality and safety of prepared dishes and the possible adverse consequences to their health when eating prepared dishes. Consumers may have different types of risk perceptions of prepared dishes in the context of asymmetric market information, varying consumer information acquisition capabilities, and risk preferences. Risk perception may play a mediating role between cognition and purchasing intention. Wang and Gao [58] found that risk perception plays a mediating role between information search and purchasing intention for consumers buying fresh products online. Bi [59] discovered that risk perception plays a mediating effect between consumers’ relationship strength and purchasing intention, with relationship strength representing the frequency and intensity of information search communication, and information exchange between parties. Xiang and Guo [60] empirically found that risk perception plays a mediating effect between financial literacy and consumers’ online loan purchase decisions. Liu and Yang [61] empirically found that risk perception plays a mediating role between lenient return policies and purchasing intention. Based on the above analyses, the following hypotheses are proposed:

**H4.** *Risk perception plays a mediating role between the effects of individual power on urban residents’ continuous purchasing intention of prepared dishes*.

**H5.** *Risk perception plays a mediating role between the effects of geographical power on urban residents’ continuous purchasing intention of prepared dishes*.

**H6.** *Risk perception plays a mediating role between the effects of cultural power on urban residents’ continuous purchasing intention of prepared dishes*.

## 3. Research Methods and Data

### 3.1. Questionnaire Design

The questionnaire is divided into three sections. The first section is to measure consumers’ continuous purchasing intention. The measurement of continuous purchasing intention refers to Zhang et al. [39]. The second section measures consumers’ individual power, geographical power, cultural power, and risk perception. The scale measurement of individual power refers to Zou et al. [47], Han [46], Jin [49], and Huang et al. [62]. For the geographical power scale, it refers to Zou et al. [47], Wang et al. [63], Sheng et al. [48], Duan [52], and Huang et al. [62]. The scale of the cultural power measurement is modified from the scale of Zou et al. [47]. The risk perception measurement refers to Zhang et al. [39]. The third section covers the demographic characteristics of consumers (including gender, age, income, education level, marital status, family structure, etc.). The questionnaire design drew on numerous related studies and involved evaluations by relevant experts, and finally formed the variable measurement table shown in Table 1, with all variables measured using a five-point Likert scale.

### 3.2. Methods and Data Analysis

This paper utilized the internationally recognized research platform *Credamo* to collect questionnaires from April to May 2024.

Before distributing questionnaires, we imposed some restrictions on respondents to enhance data quality. First, a historical response rate of over 80% was set to reduce the incidence of low-quality responses from less serious participants. Second, validations such as sliding and identity verification were implemented before respondents could answer, to prevent automated responses. Third, a distance restriction was imposed, preventing repeated responses within a 1 km radius to avoid sample homogeneity due to respondent aggregation. Fourth, consumers from the first-tier cities of Shanghai and Beijing were selected as representatives to focus on questions related to prepared dishes. In the era of advanced internet information, with China’s internet user base reaching 1.092 billion, online questionnaires are both feasible and representative. After the modification and adjustment of the questionnaire in the pre-survey, 599 questionnaires were distributed in the formal survey, and 462 valid questionnaires were obtained by excluding invalid and poor quality questionnaires, with a questionnaire survey validity rate of 77.13%. This aligns with Bentler’s recommendation that the sample size should be 10–15 times the number of observed variables [64] and meets the suggested sample size of 400–500 for structural equation modeling [65]. The sample distribution is consistent with the consumer portrait of prepared dishes, ensuring good representation.

The data analysis is performed using the software of Excel and Smart PLS 4.0. Excel is utilized for data formatting and cleaning to prepare the data for analysis. Smart PLS is employed primarily for Partial Least Squares Structural Equation Modeling (PLS-SEM) and bootstrapping functions.

PLS-SEM is used to model and analyze the reliability and validity of the model. Based on research related to Smart PLS, this study uses three indexes (Cronbach’s Alpha, CR(rh0_a), and CR(rh0_c)) to assess reliability. The critical thresholds for these indices are 0.6, 0.7, and 0.8, respectively, and exceeding these values indicates high internal consistency. Convergent validity is assessed using Average Variance Extracted (AVE), which evaluates the total amount of variance that can be explained by the latent variable relative to the measurement error. An AVE value above 0.5 signifies good convergent validity.

Discriminant validity is the idea that there should be a clear distinction between different traits and connotation measurements [66]. It is typically measured using the Fornell–Larcker criterion and the Heterotrait–Monotrait ratio (HTMT) criterion. According to the Fornell–Larcker criterion, if the square root of the AVE is greater than the correlation coefficients between the latent variable and other variables, the scale exhibits good discriminant validity [67]. According to the HTMT criterion, if the value between two latent variables is less than 0.9, the discriminant validity is considered good [68].

The hypotheses’ path coefficients are analyzed through a Bootstrapping setup with repetitive sampling for 5000 times. According to the mediation effect testing and analysis procedures by Zhao et al. [69], if the 95% confidence interval does not contain zero, the mediating effect is considered significant. The product of the indirect path coefficients and the direct path coefficients is greater than zero, indicating that the mediating variable B plays a partial mediating role in the effect of the explanatory variable A on the be explained variable C. It also indicates the presence of other mediating variables acting in the same direction as the hypothesized mediating effect. If the direct effect of the explanatory variable A’ on the be explained variable C is not significant, the indirect effect is significant and its path coefficient is positive, this indicates that the mediating variable B plays a fully mediating role in the effect of the explanatory variable A’ on be explained variable C.

## 4. Results

### 4.1. Demographic Characteristics

According to demographic characteristics, the majority of respondents are female (69.7%), reflecting that women are the primary purchasers in households, consistent with iMedia Consulting’s 2022 [70] profile of prepared dishes consumers and the finding that over 60% of household consumption decisions in China are made by women [71]. The age distribution is concentrated between 18 and 40 years old (86.2%), mainly young consumers, which may be due to the fact that young consumers are more curious about new things and have a higher demand for the convenience of prepared dishes, mirroring findings from related studies [72]. Over 86% of respondents have a bachelor’s degree or higher, and their monthly household income is evenly distributed at above 5000 RMB. A high education level and a high income level are the commonalities of these consumers. Marital status is evenly distributed, in line with the specifics of first-tier cities. In terms of family structure, 67.32% of the households have elderly people over 60 years old or children under 18 years old. The demographic characteristics of the sample are presented in Table 2.

### 4.2. Reliability and Validity Analysis

#### 4.2.1. Reliability and Convergent Validity Tests

The reliability and convergent validity test results are presented in Table 3. The reliability results of all three values exceed the critical thresholds of 0.6, 0.7, and 0.8, respectively. This indicates that the scale has a high level of internal consistency.

To enhance the reliability and validity, the measurement items of individual power and geographical power with loading factors below 0.5 were deleted. As shown in Table 3, the AVE values all exceed the critical threshold of 0.5, indicating having a good convergent validity.

#### 4.2.2. Discriminant Validity Test

The results of the Fornell–Larcker Criterion discriminant validity test, as shown in Table 4, the square root of the AVE exceeds the correlation coefficients between each latent variable and other variables. This indicates that the scale has a good discriminant validity.

The HTMT discriminant validity test results all fall below 0.9, as shown in Table 5, demonstrating that the variables in this study exhibit good discriminant validity.

### 4.3. Hypothesis Testing

The standardized path coefficients are plotted as shown in Figure 2. The hypotheses presented in the previous section are tested based on the path coefficient results and their significance. The results show that all hypotheses are valid except H3, and the standardized coefficients pass the test at 0.001 significance level.

#### 4.3.1. Direct Effects Analysis

Model hypotheses’ path coefficients for direct effects are shown in Table 6.

Individual power has a significant positive effect on the intention to continue purchasing, which verifies H1. The path coefficient of the effect of individual power on continuous purchasing intention is 0.440, indicating its important role in the continuous purchasing intention of prepared dishes. The more information residents obtain or use from these channels to form their own cognition about prepared dishes, the more importantly they influence their continuous purchasing intention.

Geographical power has a significant positive effect on the intention to continue purchasing, which verifies H2. The path coefficient of the effect of geographical power on continuous purchasing intention is 0.276. The deeper the cognition of prepared dishes, the higher the matching of demand and the stronger the intention of continuous purchasing.

However, cultural power has no significant direct effect on the continuous purchasing intention, thereby rejecting H3.

#### 4.3.2. Mediating Effects Analysis

Model indirect effects testing results are shown in Table 7. The indirect path coefficient of individual power on consumers’ continuous purchasing intention is 0.052, which is smaller than its direct path coefficient of 0.440, and H4 is verified. The product of the indirect path coefficients and the direct path coefficients of individual power is greater than zero, indicating that risk perception plays a partial mediating role in the effect of individual power on continuous purchasing intention. The mediating effect indicates that risk perception plays a negative mediating role in the influence of individual power on consumers’ continuous purchasing intention of prepared dishes. Residents’ experiences and preferences in purchasing prepared dishes form their judgment of the risk level of prepared dishes. Higher perceived risk among residents correlates with lower intention to continue purchasing prepared dishes.

Similarly, the indirect path coefficient of geographical power is 0.046, which is smaller than its direct path coefficient of 0.276, and H5 is verified. The product of the indirect path coefficients and the direct path coefficients of geographical power is greater than zero, indicating that risk perception plays a partial mediating role in the effect of geographical power on continuous purchasing intention. Risk perception plays a negative mediating role in the influence of geographical power on consumers’ continuous purchasing intention for prepared dishes, indicating that the negative evaluation of prepared dishes by the people around the consumers raises their risk perception, which then reduces the continuous purchasing intention of prepared dishes.

The product of the indirect path coefficients and the direct path coefficients of individual and geographical power is greater than zero, indicating that research about residents’ continuous purchasing intention for prepared dishes has other mediating variables in the model.

Since the direct effect of cultural power on continuous purchasing intention is not significant, the indirect effect is significant and its path coefficient is 0.048, indicating that risk perception plays a fully mediating role in the effect of cultural power on continuous purchasing intention, and H6 is verified. Risk perception plays a negative mediating role in cultural power and continuous purchasing intention. It indicates that consumers learn more negative news and events of government monitoring about prepared dishes in the market and then they will have a strong risk perception about them, which reduces consumers’ continuous purchasing intention.

## 5. Discussion

In this paper, by introducing the distributed cognition theory and using risk perception as a mediating variable, we explore the mechanisms influencing consumers’ continuous purchasing intention of prepared dishes from different dimensions of cognition. Specifically:

Firstly, this study confirms that distributed cognition theory provides a critical theoretical foundation for explaining consumers’ continuous purchasing intention in prepared dishes markets. Previous studies often relied on theories such as the Theory of Planned Behavior to study food purchasing intention. This study broadens the application of distributed cognition theory and deepens the research on the field of prepared dish consumption.

Secondly, cognition directly or indirectly impacts residents’ continuous purchasing intention, aligning with the findings of Zheng et al. [73], who emphasized the significant role of cognition in shaping consumers’ continuous purchasing intention. In terms of individual power, as residents accumulate relevant knowledge in the process of purchasing prepared dishes, they develop a deeper understanding of the different brands, flavors, and nutritional content of prepared dishes. This accumulation of knowledge enables consumers to make more wise and suitable purchasing decisions, thereby enhancing their purchasing intention for prepared dishes. Consumers gather information about prepared dishes through various channels, such as shopping websites and micro-blogs, which provide extensive product details as well as consumer reviews and feedback. By actively utilizing these resources, consumers compare the advantages and disadvantages of different products, form preferences, and establish judgment criteria. This ability to acquire information allows them to have a more comprehensive understanding of prepared dishes, which in turn enhances their purchasing intention.

Geographical power, an important part of an individual’s social circle, also significantly influences purchasing decisions through the comments and opinions of family members, relatives, and friends. Positive feedback from these sources enhance trust and purchasing intention, whereas negative comments may deter purchasing intention. Consumers are more inclined to purchase prepared dishes when they perceive that such consumption aligns with current social trends emphasizing fashion, convenience, and health. This alignment may be driven by consumer desires for modern lifestyles or influenced by external factors like media and online opinions. These findings are consistent with studies on GM foods [74] and organic milk [75], where knowledge and purchasing intention are interconnected. Furthermore, peer effects and herd behavior demonstrate that individuals with greater personal experience and understanding, tend to make more rational consumption choices based on others’ evaluations. In contrast, those lacking product knowledge are more susceptible to the influence of their social circles [41].

Cultural power, a multifaceted factor influencing consumers’ purchasing intention, encompasses knowledge about the complaint process, government departmental regulation, and legal norms regarding the safety of prepared dishes, as well as perceptions of the handling of specific incidents (e.g., the use of feet in Chinese sauerkraut production, issues in school cafeterias, and incidents involving substandard pork in prepared dishes). Cultural power does not significantly impact the continuous purchasing intention of prepared dishes, it may be due to several reasons: (1) The level of cognition regarding the complaint process, regulatory oversight, and legal standards varies among consumers. Some are well-informed and highly concerned, while others lack knowledge. (2) Although the prepared dishes safety incidents have aroused public concern to a certain extent, their direct impact on consumers’ purchasing intention may be limited. On the one hand, the exposure and handling of the incident may have heightened consumers’ concerns about the safety of prepared dishes, but on the other hand, consumers may have been more concerned about factors directly related to their purchasing decisions. In addition, this study’s findings have a contrast with those of Zou et al. [47], which focused on rural residents’ food safety consumption. That study found that individual power and geographical power did not directly influence continuous purchasing intention, whereas cultural power did. These discrepancies may stem from differences in economic levels between urban and rural areas, consumer awareness, information and purchasing channels, and the extent of governmental focus [76], leading to varying study outcomes.

Thirdly, risk perception plays a mediating role in the effect of cognition on continuous purchasing intention. Specifically, it partially mediates the effects of individual power and geographical power on continuous purchasing intention, consistent with the findings of Cheng and Yin [77]. The key components of cultural power encompass laws, regulations, and regulatory events related to prepared dishes. Residents tend to focus on negative news occurring in the prepared dishes market, such as food safety and quality issues caused by manufacturers’ ethical problems, as well as legal regulation and complaint feedback process of prepared dishes. When government actions related to these events are clear and well-communicated, consumers’ risk perception will decrease, thereby increasing their intention to purchase prepared dishes continuously. This observation aligns with Liu and Yang [61], who found that risk perception mediates the relationship between policy leniency and purchase intention. It also broadly concurs with Zou et al. [47], except that risk perception partially mediates the effect of cultural power on purchasing intention. This discrepancy might arise from differences in consumer focus between general food safety and the continuous purchasing of prepared dishes. In the former case, government norms and authoritative information have a direct impact, while in the latter, the reduction in risk perception due to improved governmental oversight enhances continuous purchasing intention.

## 6. Conclusions and Managerial Implications

This paper takes urban residents in Shanghai and Beijing as the object, combines distributed cognition theory, and analyzes the influence paths of cognition in the three dimensions of individual power, geographical power, and cultural power on urban residents’ continuous purchasing intention of prepared dishes. The conclusions are as follows:

Individual power has the greatest impact on residents’ continuous purchasing intention of prepared dishes. In the process of purchasing prepared dishes, consumers can obtain information from several channels. These channels not only increase their knowledge of prepared dishes products but also directly affect their purchasing intention. Shopping websites provide detailed information for easy comparison and selection, and promotional offers enhance the desire to buy. Food bloggers on microblog share their experience of using prepared dishes as well as discuss with users to form a word-of-mouth effect, which boosts purchasing intention. Crucially, when prepared dishes accurately match individual dietary preferences, they not only satisfy the taste buds but also save time, which greatly enhances their purchasing intention. Therefore, firstly, to cultivate and enhance consumer cognition, enterprises need to objectively and truthfully disclose product information regarding the production process, nutritional labels, and test results. These will help consumers comprehensively understand the product during the purchasing process. Secondly, enterprises should make more efforts to publicize the functions and knowledge of prepared dishes, enhancing consumers’ cognition and catering to the different consumption needs of different residents. Thirdly, in response to the asymmetry of market information, enterprises need to improve the information flow channels, with the help of the internet platform to actively and objectively publicize the information about prepared dishes, and reduce consumer doubts about the purchasing of prepared dishes.

The influence of geographical power on the residents’ continuous purchasing intention of prepared dishes cannot be ignored. Consumers can obtain direct feedback on prepared dishes from reviews of family members, relatives, and friends, which are often based on actual experiences, including taste, convenience, and nutritional value. Positive reviews can enhance consumers’ trust and favorability towards prepared dishes, which in turn boosts purchasing intention. Conversely, negative reviews may cause doubts and deter purchasing. Meanwhile, consumers’ personal economic considerations are one of the most important factors in determining purchasing intention. When consumers perceive that the price of prepared dishes is within their acceptable range, this will reduce the financial pressure at the time of purchase and increase the attractiveness of the purchase. In addition, social consumption trends have a non-ignorable impact on individual purchasing behavior. When consumers perceive that purchasing prepared dishes is in line with the current social consumption trend of pursuing convenience, efficiency, and health, this perception can motivate them to purchase these products as part of a fashionable lifestyle. Therefore, firstly, from the perspective of consumers’ cognition of geographical power, enterprises should pay attention to the peer effect and information transmission among consumers. By enhancing product and service quality, enterprises can build a positive reputation and value through word-of-mouth, facilitating consumer recognition and acceptance of prepared dishes. Secondly, residents’ food choices are influenced by conformity effects. Under the prevalent trend of pursuing popular dining experiences, residents seek dietary concepts that align with contemporary trends. To improve consumers’ cognition of this new type of prepared dishes, enterprises can engage celebrities and internet influencers to endorse their products, leveraging the demonstration effect of celebrities and public figures to enhance consumers’ continuous purchasing intention of prepared dishes.

Cultural power, as an intangible aspect of the social environment encompassing culture and institutions, indirectly influences purchasing intention, indicating that residents are very concerned about food safety. Although cultural power does not have a significant direct effect on continuous purchasing intention, it influences residents’ continuous purchasing intention through the mediating role of risk perception. Consumers are concerned about the loss of nutrients, excessive additives, and food safety issues in prepared dishes, which affect their healthy choices. Additionally, doubts regarding taste, freshness, and shelf life also diminish purchasing interest. Therefore, to enhance consumer purchasing intention, it is crucial to ensure that prepared dishes maintain balanced nutrition, contain safe additives, meet food safety standards, and offer improved taste and freshness. Therefore, firstly, prepared dish enterprises need to strictly abide by the industry norms, comply with the industry ethics, uphold the concept of consumer-centered, and firmly guard the insurmountable bottom line of food safety to guarantee the safety and quality of prepared dishes. Secondly, the government needs to strengthen the formulation and dissemination of relevant regulations, intensify law enforcement against illegal prepared dishes enterprises, and improve and publicize the relevant laws, regulations, and complaint channels in a timely and transparent manner, so as to create a favorable environment for residents to consume prepared dishes.

## 7. Limitations and Future Research

First, this paper examines the effect of cognition on residents’ continuous purchasing intention of prepared dishes, considering the mediating role of risk perception. The mechanism of cognition’s influence on the purchasing intention of prepared dishes is a complex and multidimensional process, and the mechanism of the role of emotional factors in the cognitive process, as well as how social interactions affect individuals’ cognition and purchasing decisions need further detailed research. Therefore, in the future, multidisciplinary theories and methods should be comprehensively applied, and other suitable mediating and moderating variables should be induced to explore the mechanism of cognition’s influence on the consumption intention of prepared dishes from more perspectives, so as to provide a more comprehensive and deeper insight for the understanding of purchasing intention and behavior of prepared dishes.

Secondly, this paper focuses on the purchasing intention of prepared dishes but does not analyze the purchasing behavior. Although some studies have shown a strong positive relationship between purchasing intention and behavior, more practical contextual and external factors need to be considered in order to transform this willingness into actual purchasing behavior. Purchasing behavior is influenced not only by consumers’ individual cognition but also by a variety of factors such as market environment, price sensitivity, purchasing convenience, social norms, laws, and regulations. For example, consumers may adjust their purchasing behavior due to external factors such as price changes, promotional activities, and product stock-outs, which may not have been adequately taken into account in the formation of purchasing intention. Therefore, future research could further explore the specific mechanisms of transformation between purchasing intention and behavior of prepared dishes, including which factors act as a bridge between the two and how these factors interact with each other to influence the final purchase behavior.

Finally, future research should categorize the types of prepared dishes, such as ready-to-eat, ready-to-heat, ready-to-cook, ready-to-serve, etc., to explore consumer preferences, cognitive formation processes, and characteristics of purchasing behavior under different categories. In addition, the influence of factors such as regional culture and eating habits on the cognition of prepared dishes needs to be considered to construct a more comprehensive model of consumer behavior. Through refined analysis, enterprises can more accurately locate the target market, optimize product design and marketing strategies, meet diversified consumer demands, and promote the healthy development of the prepared dishes industry.

## Figures and Tables

**Figure 1 foods-13-02598-f001:**
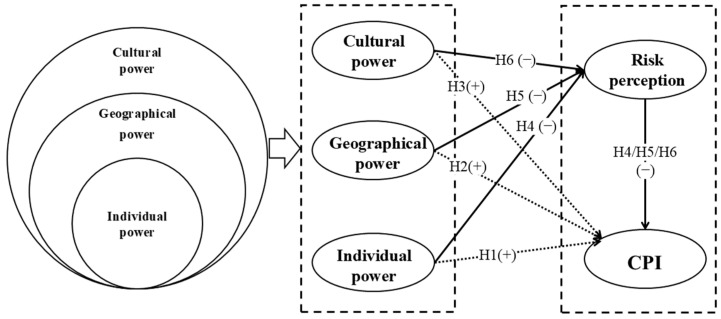
Structural equation model.

**Figure 2 foods-13-02598-f002:**
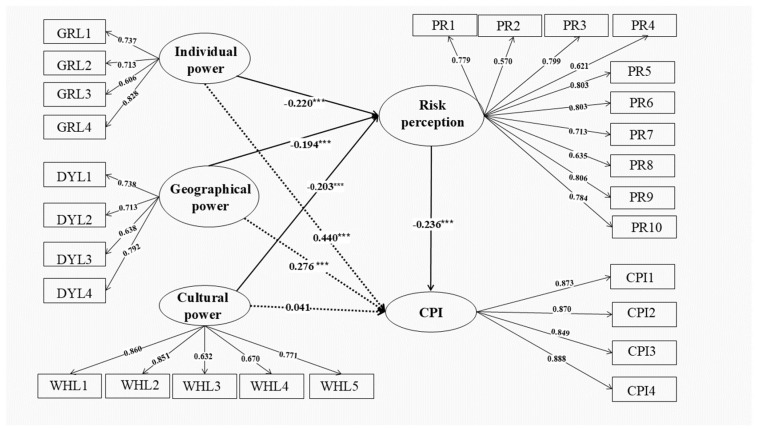
Structural equation modeling results. *** represents *p*-values significance at the 0.001 level.

**Table 1 foods-13-02598-t001:** Variable Measurement Scale.

Variable Category	Latent Variable	Observed Variable
Explanatory variable	Individual power	Knowledge gained from purchasing prepared dishes (GRL1)
		Knowledge of prepared dishes from shopping websites (GRL2)
		Knowledge of Prepared Dishes gained from Micro-blog (GRL3)
		Buying prepared dishes bases on my dietary preference (GRL4)
	Geographical power	Knowledge gained from family consumption evaluations of prepared dishes (DYL1)
		Knowledge gained from relatives’ and friends’ evaluations of prepared dishes (DYL2)
		The price of prepared dishes is acceptable (DYL3)
		Perception of prepared dishes comes from the consuming trend of the society (DYL4)
	Cultural power	Knowledge degree of the complaints process about the safety issues of prepared dishes (WHL1)
		Level of knowledge of legal norms and government departments regulating prepared dishes (WHL2)
		Level of understanding of government departments handle “using feet in Chinese sauerkraut making process” (WHL3)
		Level of understanding of government departments handle “prepared dishes going to school” (WHL4)
		Level of understanding of government departments handle “prepared dishes cooked with poor-quality pork” (WHL5)
Mediate variable	Risk perception	I concern the quality and safety of prepared dishes, and eating them may be harmful to my health (PR1)
		Eating questionable prepared dishes will increase my medical expenses (PR2)
		Eating prepared dishes is similar to takeout and is bad for my health in the long run (PR3)
		Prepared dishes are closely related to health, I must buy cautiously (PR4)
		I concern excessive or illegal food additives and preservatives in prepared dishes (PR5)
		I concern the use of substandard raw materials for prepared dishes (PR6)
		Prepared dishes have a lot less nutritional value than traditionally cooked dishes (PR7)
		The flavor and texture of prepared dishes are worse than traditionally cooked dishes (PR8)
		Packaging materials for prepared dishes are of varying quality and may produce harmful substances (PR9)
		Difficulty in ensuring transportation conditions (e.g., refrigerated, frozen, etc.) for prepared dishes during transportation (PR10)
Explained variable	Continuous purchasing intention	In the future, I would like to continue to buy prepared dishes (CPI1)
		I would recommend prepared dishes to friends and family around me. (CPI2)
		I would like to prioritize cooking with prepared dishes over traditional cooking methods in the future (CPI3)
		If I had to choose again, I’d still prefer to cook with prepared dishes (CPI4)

Questionnaire presents in Chinese when it been distributed.

**Table 2 foods-13-02598-t002:** Descriptive statistics (*N* = 462).

Variable	Categorization	Frequency	Percentage/%
Gender	Male	140	30.30
	Female	322	69.70
Age	18–30	236	51.08
	31–40	162	35.06
	41–50	44	9.52
	51–60	22	4.76
	else	2	0.43
Educational level	Junior high school and below	4	0.87
	Senior high school	16	3.46
	Junior college	41	8.87
	Undergraduate	286	61.90
	Postgraduate and above	115	24.89
Average monthly household income	Less than 5000 RMB	21	4.55
	5001–10,000 RMB	107	23.16
	10,001–15,000 RMB	98	21.21
	15,001–20,000 RMB	108	23.38
	More than 20,000 RMB	128	27.71
Marital status	Spinsterhood	230	49.78
	Married	232	50.22
Family structure	No elderly and child	151	32.68
	With a senior citizen over 60 years old or a child under 18 years old	311	67.32

1 dollar ≈ 7.25 RMB, 1 euro ≈ 7.87 RMB, 5000 RMB ≈ 635.26 euro ≈ 689.56 dollars.

**Table 3 foods-13-02598-t003:** Reliability and convergent validity test.

Indicators	Items	Normalized Factor	Cronbach’s Alpha	CR(rh0_a)	CR(rh0_c)	AVE
Individual power	GRL1	0.749	0.708	0.768	0.814	0.526
	GRL2	0.719				
	GRL3	0.598				
	GRL4	0.820				
Geographical power	DYL1	0.735	0.701	0.726	0.813	0.522
	DYL4	0.773				
	DYL2	0.721				
	DYL3	0.659				
Cultural power	WHL1	0.864	0.824	0.881	0.872	0.581
	WHL2	0.865				
	WHL3	0.614				
	WHL4	0.658				
	WHL5	0.764				
Risk perception	PR1	0.779	0.904	0.916	0.921	0.542
	PR2	0.571				
	PR3	0.798				
	PR4	0.621				
	PR5	0.803				
	PR6	0.803				
	PR7	0.713				
	PR8	0.635				
	PR9	0.807				
	PR10	0.784				
Continuous purchasing intention	CPI1	0.868	0.893	0.894	0.926	0.757
	CPI2	0.869				
	CPI3	0.853				
	CPI4	0.891				

**Table 4 foods-13-02598-t004:** Fornell-Larcker criterion discriminant validity test results.

	CPI	DYL	GRL	PR
CPI	0.870			
DYL	0.684	0.722		
GRL	0.747	0.640	0.725	
PR	−0.576	−0.443	−0.457	0.736
WHL	0.532	0.533	0.552	−0.428

**Table 5 foods-13-02598-t005:** Heterotrait–Monotrait ratio (HTMT) discriminant validity test results.

	CPI	DYL	GRL	PR
CPI				
DYL	0.820			
GRL	0.879	0.864		
PR	0.625	0.524	0.524	
WHL	0.582	0.667	0.683	0.444

**Table 6 foods-13-02598-t006:** Model hypotheses-path coefficients for direct effects.

Hypothesis	Path	Path Coefficient	Sample Mean (M)	STDEV	T	*p*	Results
H1	GRL- > CPI	0.440	0.440	0.035	12.600	0.000	Support
H2	DYL- > CPI	0.276	0.276	0.035	7.931	0.000	Support
H3	WHL- > CPI	0.041	0.042	0.035	1.149	0.250	Refuse

**Table 7 foods-13-02598-t007:** Model indirect effects testing results.

Hypothesis	Path	Original Sample (O)	Sample Mean (M)	STDEV	T	*p*	Results
H4	GRL- > PR- > CPI	0.052	0.052	0.012	4.365	0.000	Support
H5	DYL- > PR- > CPI	0.046	0.046	0.013	3.643	0.000	Support
H6	WHL- > PR- > CPI	0.048	0.048	0.012	3.924	0.000	Support

## Data Availability

The data that support the findings of this study are available from the corresponding authors upon reasonable request.

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
