# Peer review of "How Cognition Influences Chinese Residents’ Continuous Purchasing Intention of Prepared Dishes under the Distributed Cognitive Perspective"

_foods, 2024, doi:10.3390/foods13162598_

Round 1

Reviewer 1 Report

Comments and Suggestions for Authors

In the introduction, the authors present the context of the research, focusing mainly on China. In my opinion, this is not a good approach. It should start with the international context, regional (Asia) and then move on to the national context. An overview of the research questions would be a better introduction to the topic. Also, even in this introductory part, the authors should cite recent literature sources.

The authors identify the gap in the literature very well, but the literature review is not well done and again cites too many Chinese authors or studies. The literature review should include international studies. In the methodology section the authors directly present the research instrument (questionnaire). There is no prior description of the research method (survey). The demographic profile of respondents should be presented in the results section, not in the methodology. Also, it is not clear from the methodology what the data collection period was.

The software with which the data was analyzed is not mentioned.

The section dedicated to the presentation of results is too short and superficial compared to the other sections. The discussions can be improved: the results of the present study should be compared with those of other studies in the international literature (if the authors improve the introduction and the literature review, they will be able to successfully improve this section as well).

The policy implications are not sufficiently developed.

Author Response

Comments 1: In the introduction, the authors present the context of the research, focusing mainly on China. In my opinion, this is not a good approach. It should start with the international context, regional (Asia) and then move on to the national context. An overview of the research questions would be a better introduction to the topic. Also, even in this introductory part, the authors should cite recent literature sources.

Response 1: Thank you for pointing it out. We agree with this comment. Therefore,

Through searching other international literature, we added some literature in literature review part. (Due to the revision, we cited reference in the lines 61-75)

Comments 2: The authors identify the gap in the literature very well, but the literature review is not well done and again cites too many Chinese authors or studies. The literature review should include international studies. In the methodology section the authors directly present the research instrument (questionnaire). There is no prior description of the research method (survey). The demographic profile of respondents should be presented in the results section, not in the methodology. Also, it is not clear from the methodology what the data collection period was.

Response 2: We agree with your comment. So according to your suggestion, we have re-written this part. Briefly speaking, we:

(1) Added some international literature. (In page 2, paragraph 3, line 61-75)

(2) Re-wrote the part 3.2.Methods and data analysis, integrating other parts theoretical analysis into methods analysis. (In page 7, paragraph 2-4, line 244-272)

(3) Put the demographic profile of respondents in the part 4.Results. (In page 7, paragraph 5, line 274-292)

(4) Added the data collection period in part 3.2.(In page 6, paragraph 1, line 226)

Comments 3: The software with which the data was analyzed is not mentioned.

Response 3: Thank you very much for your detailed guidance. According to your suggestion, we have added this part. Briefly speaking, we:

Added that we used data analysis software of Excel and Smart PLS 4.0. (In page 7, paragraph 2, line 244)

Comments 4: The section dedicated to the presentation of results is too short and superficial compared to the other sections. The discussions can be improved: the results of the present study should be compared with those of other studies in the international literature (if the authors improve the introduction and the literature review, they will be able to successfully improve this section as well).

Response 4: As you suggested, we re-wrote the results and conclusions part to explain deeply about our topic. (Results In page 8-11, line 294-354); (Conclusions In page 13-14, line 455-506)

Comments 5: The policy implications are not sufficiently developed.

Response 5: Thank you very much for your suggestion. As you suggested, we have re-written our conclusions part, and after the conclusion statement of every paragraph is policy implication(or managerial implication). There we have not modified it.(In page 13-14, line 463-472,487-496,506-514)

Thank you very much for your time and valuable suggestions!

Reviewer 2 Report

Comments and Suggestions for Authors

Abstract. The abstract is written clearly and encompasses all the necessary elements. However, I suggest clarifying the main purpose of the research and outlining main contributions to the body of knowledge.

Introduction. The research background is described and previous research in the field is presented clearly. The gap in the literature is revealed. However, the aim statement is omitted (the statement “this paper aims to offer some theoretical support to enterprises in developing marketing programs related to prepared dishes and government departments in refining relevant policies” cannot be considered as the aim). Also, the research questions are disputable. For example, if the answer to the first part of the question was “No”, why would the second part of this question be necessary?

Research Hypothesis. First, as more than one hypothesis is provided, the title should be provided in plural (Hypotheses). The hypotheses are formulated properly and substantiated by theory. Also, I suggest including the hypothetical model at the end of this section.

Research Methods and Data. The formation of the questionnaire is properly explained. However, I doubt the statements. E.g., statements “Knowledge gained from purchasing prepared dishes (GRL1)” and “Buying prepared dishes matches my dietary preference (GRL4)” cannot be assessed using the same scale, because in GRL1 one should evaluate the level of knowledge, and in GRL4 – the level of agreement with the statement. The language of the questionnaire was not indicated. The same incompatibilities also occur with other statements.

While presenting income, the Authors should provide the currency equivalent in USD or EUR for the researchers from other countries to get perception of the amount (this could be provided as a note under the table).  Also, the country’s average income could also be indicated.

Result. In my opinion, Reliability analysis is not a result. I suggest providing it as a substantiation for the questionnaire in the method-related section of the paper.

Also, it is not good providing Hypothesis Testing as a sub-section of the reliability analysis. Hypothesis testing is the main result of the paper. As there was no hypothetical model provided, the explanation of H3 should be provided in lines 262-263: Cultural power has no significant effect on urban residents continuous purchasing intention of prepared dishes.

Discussion. Discussion is provided properly; however, I would suggest relating it to hypotheses (with special attention to H3) more tightly.

Conclusions. If the section 6 is already called “Conclusions”, calling the same 6.1 is not a good idea. Also, one sentence for the future research is not enough.

Author Response

Comments 1: Abstract.The abstract is written clearly and encompasses all the necessary elements. However, I suggest clarifying the main purpose of the research and outlining main contributions to the body of knowledge.

Response 1: Thank you for pointing this out. We agree with your comment. Therefore,

Through referring to other international literature and your suggestion,we modified our research purpose(In page 1, Abstract, lines 11-12), and the contributions have already existed(In page 3, paragraph 4, lines 109-115).

Response 2: Thank you for pointing these problems out. We agree with your partial comments. Therefore,

(1)According to the your suggestion, I think a timeline of the background would be helpful for understanding this article,we can not to modify it. And we added some background.(In page 1, paragraph 1, lines 43-46)

(2) As we discuss it, we recognize the omission of the purpose statement.(In page 3, paragraph 4, lines 104-109)

(3) Our research questions are a juxtaposition and represent two paths of the model. The first question has no logical effect on the second question.

Comments 3: Research Hypothesis. First, as more than one hypothesis is provided, the title should be provided in plural (Hypotheses). The hypotheses are formulated properly and substantiated by theory. Also, I suggest including the hypothetical model at the end of this section.

Response 3: Thank you very much for pointing the details out. We agree with your comments. Therefore, According to the your suggestion,

1. We modified the”hypotheses”in the lines 121.

2. We added the Figure 1. Due to the revision, now it is located in page 5, line 205,like this:

Comments 4: Research Methods and Data. The formation of the questionnaire is properly explained. However, I doubt the statements. E.g., statements “Knowledge gained from purchasing prepared dishes (GRL1)” and “Buying prepared dishes matches my dietary preference (GRL4)” cannot be assessed using the same scale, because in GRL1 one should evaluate the level of knowledge, and in GRL4 – the level of agreement with the statement. The language of the questionnaire was not indicated. The same incompatibilities also occur with other statements.While presenting income, the Authors should provide the currency equivalent in USD or EUR for the researchers from other countries to get perception of the amount (this could be provided as a note under the table).  Also, the country’s average income could also be indicated.

Response 4: Thank you for pointing out these questions. We are so sorry to translate with wrong because English isn’t our mother tongue. And I admit that the question design should use different scales, but I think “dietary preference” also can be considered knowledge source. So according to the your suggestion, we have re-written this part. Briefly speaking, we:

(1) Modified the statement of GRL4. (In page 5, Table 1)

(2) Examined other questions statement and adjusted a little .(In page 5, Table 1)

(3) Noted the currency equivalent in USD and EUR with RMB under the Table 2. But we think Shanghai and Beijing is the China most rich two cities, the country’s average income is short of value to compare with our data. (In page 8, paragraph 2, line 300).

Comments 5: Result. In my opinion, Reliability analysis is not a result. I suggest providing it as a substantiation for the questionnaire in the method-related section of the paper.Also, it is not good providing Hypothesis Testing as a sub-section of the reliability analysis. Hypothesis testing is the main result of the paper. As there was no hypothetical model provided, the explanation of H3 should be provided in lines 262-263: Cultural power has no significant effect on urban residents continuous purchasing intention of prepared dishes.

Response 5: Thank you for your suggestion. Through our literature research, we find our paper’s problem in the reliability analysis part. Therefore, we integrated all theoretical analysis of methods in part 3.2. And Hypothesis testing part have modified. Briefly speaking, we:

(1) Added the analysis of methods. (In page 7, paragraph 2, line 248-278),

(2) Modified to 4.3 Hypothesis testing.(In page 10, paragraph 2, line 323).

Response 6: Thank you for your suggestion. We agree with your comments. Therefore,

we divided the paragraph”firstly,......”into three paragraphs to state in detail.(In page 12, paragraph 1-3, line 386-424)

Response 7: According to the your suggestion, we

(1) Modified the title from 6.1 to 6. Conclusions and Managerial Implications(In page 13,line 449), and modified title 6.2 to 7. Limitations and Future Research. (In page 14, line 515)

(2) Expanded the limitations and future research part. (In page 14-15, paragraph 3-5, line 515-549)

Thank you very much for your time and valuable suggestions!

Round 2

Reviewer 1 Report

Comments and Suggestions for Authors

Dear authors,

Congratulations! The paper was significantly improved after the first round of revision.

Author Response

Comments 1: Congratulations! The paper was significantly improved after the first round of revision.

Response 1: Thank you very much. It is with your valuable suggestions that we can find the deficiency of our paper and improve it.

Reviewer 2 Report

Comments and Suggestions for Authors

Dear Authors, thank you for considering my comments. I have some more considerations:

1) Please, indicate the language used in the survey (questionnaire). This should be done in the section 3.1.

2) Please, translate the model "Figure 2. Structural equation modeling results" to English languages.

Good luck in your future research.

Author Response

Comments 1:Please, indicate the language used in the survey (questionnaire). This should be done in the section 3.1.

Response 1:Thank you for your suggestions. As you suggested, we write a note under the Table1.(In the page 6,line 233)

Comments 2:Please, translate the model "Figure 2. Structural equation modeling results" to English languages.

Response 2:Thank you for pointing it out. I am so sorry to make a mistake in here.The figure 2 has been translated in English.(In the page 10)

Thank you very much for your time and valuable suggestions!